# Protocol for the Pathways Study: a realist evaluation of staff social ties and communication in the delivery of neonatal care in Kenya

Conrad Wanyama [1] ,[1] Claire Blacklock,[2] Juliet Jepkosgei,[1] Mike English [1,3]
Lisa Hinton [1] ,[4] Jacob McKnight,[5,6] Sassy Molyneux,[2,7] Mwanamvua Boga,[7]
Peris Muoga Musitia [1] ,[8] Geoff Wong[9]

CW and CB contributed equally.

**Correspondence to**
Mr. Conrad Wanyama;
CWanyama@kemri-wellcome.org

## ABSTRACT

**Introduction** The informal social ties that health workers form with their colleagues influence knowledge, skills and individual and group behaviours and norms in the workplace. However, improved understanding of these 'software' aspects of the workforce (eg, relationships, norms, power) have been neglected in health systems research. In Kenya, neonatal mortality has lagged despite reductions in other age groups under 5 years. A rich understanding of workforce social ties is likely to be valuable to inform behavioural change initiatives seeking to improve quality of neonatal healthcare.
This study aims to better understand the relational components among health workers in Kenyan neonatal care areas, and how such understanding might inform the design and implementation of quality improvement interventions targeting health workers' behaviours.

**Methods and analysis** We will collect data in two phases. In phase 1, we will conduct non-participant observation of hospital staff during patient care and hospital meetings, a social network questionnaire with staff, in-depth interviews, key informant interviews and focus group discussions at two large public hospitals in Kenya. Data will be collected purposively and analysed using realist evaluation, interim analyses including thematic analysis of qualitative data and quantitative analysis of social network metrics. In phase 2, a stakeholder workshop will be held to discuss and refine phase one findings.
Study findings will help refine an evolving programme theory with recommendations used to develop theory-informed interventions targeted at enhancing quality improvement efforts in Kenyan hospitals.

**Ethics and dissemination** The study has been approved by Kenya Medical Research Institute (KEMRI/SERU/CGMR-C/241/4374) and Oxford Tropical Research Ethics Committee (OxTREC 519-22). Research findings will be shared with the sites, and disseminated in seminars, conferences and published in open-access scientific journals.

## STRENGTHS AND LIMITATIONS OF THIS STUDY

⇒ Realist evaluation is an appropriate methodology to develop middle-range programme theory, which will be useful for informing the practical design and implementation of quality improvement interventions in neonatal units.
⇒ The Pathways Study is the first to use social network analysis to explore the influence of staff social ties on the delivery and quality of neonatal care in Kenya.
⇒ Middle-range programme theory will be developed using a robust and transparent process, using an initial programme theory, involvement of stakeholders, substantive sociological theory and continual collaborative and reflexive working within a diverse research team.
⇒ The Pathways Study uses a mixed-methods approach to collect diverse data, but it is possible that some relevant data may still not be captured, particularly as some aspects of health systems 'software' are notoriously difficult to elicit.

## INTRODUCTION

Despite significant reductions in mortality rates across other age groups among children under 5 in recent years, the neonatal mortality rate in Kenya has remained relatively unchanged.[1 2] In response to this disparity, efforts are ongoing to improve the quality of neonatal care, the majority of which is delivered from public health facilities and mostly by nurses.[3] Such improvement efforts have included the recent introduction of new technologies in selected neonatal units, establishing routine neonatal data collection and feedback systems, and ongoing investment in the neonatal workforce through training and workplace support.[4 5]

However, a more detailed understanding of tacit human factors, or health systems 'software',[6] that also influence the quality of neonatal care delivered in these hospitals is now needed, to inform better focused design and implementation of improvement efforts (eg, the relationships between staff, values,

norms and the social and cognitive skills that complement clinical technical skills).[7] 'Software' in health systems (in contrast to 'hardware', such as drugs, equipment, number of staff, etc) has been defined as 'the ideas and interests, values and norms, and affinities and power that guide actions and underpin the relationships among system actors and elements'.[6] Indeed, such 'software' factors can profoundly influence acquisition of new knowledge and behaviours and norms at individual and group level,[8–11] and thus are determinants of quality care.[12 13] Moreover, individual health worker performance is shaped not only by technical clinical competencies and experiences, but also by social and cognitive skills (cognitive, social and personal resource skills such as communication, team work, leadership, situational awareness, assertiveness and decision making, coping with stress and managing fatigue, etc).[14] Relational ties among the workforce and other 'software' aspects matter greatly in the delivery of quality neonatal care, for example, care pathways, clinical guidelines, knowledge and other competencies, including those that are transferred from an educational setting such as those gained during postbasic nurse training, to a clinical care environment.[10] The impact of relational ties on clinical decision making was demonstrated in the landmark study by Coleman et al[15] showing profound peer influence on prescribing of a new drug by physicians in the USA. More recent studies have likewise demonstrated peer group influence on both clinical practices and association with differences in patient outcome.[16–18] Communication which impacts on health worker decision making may furthermore take place in 'back stage' ad-hoc opportunistic exchanges, for example, in hospital corridors,[8] which can present challenges for capture in research. Despite these challenges for researchers, understanding the ways in which health workers communicate and work with one another is vital. Better understanding of healthcare 'software' is particularly important in the neonatal unit, where patients often have multiple complex problems that require care by a multiprofessional team who are often working across different units (eg, maternity, neonatal unit, paediatrics).

Since the care of neonates is highly dependent on teams of staff working together, relational ties and social networks are likely to be central to the adoption of new or better care practices, such as those promoted in existing improvement interventions in Kenyan neonatal units. Understanding how and why communication occurs between health workers will help to unpack the many complex influences on staff behaviours and patient care. Furthermore, detailed understanding of causation with a focus on explanatory 'mechanisms' (box 1) will help to identify aspects of 'context' (box 1) amenable to intervention that are likely to result in desirable change in 'outcome' (box 1), perhaps overlooked by more traditional research approaches. Through developing explanatory middle-range programme theory (box 1), practical opportunities for improving software in neonatal care will be identified.[19 20]

## Box 1  Definitions of key terms[19 23]

**Context**
⇒ A specific aspect of the setting/environment of study, which when present triggers the activation of a mechanism(s).

**Mechanism**
⇒ A latent (often invisible) property/entity, sensitive to variations in context, which when activated causes an outcome to occur.

**Outcome**
⇒ A desirable (or undesirable) event/occurrence, which is of interest by its presence or absence (or strength/degree of).

**Context-mechanism-outcome configuration (CMOC)**
⇒ A unit/statement/diagram which links and orders a context, mechanism and outcome, to provide causal explanation. It is often abbreviated to CMO or CMOC.

**Middle-range programme theory**
⇒ The product of realist enquiry—an abstracted theory derived from relevant and robust data, which is both testable, and transferable to other settings by application of abstracted learning.

**Initial programme theory**
⇒ A rough outline of a theory of causation pertaining to the question under study, to guide the process of investigation.

**Demi-regularity**
⇒ An observable pattern in the data, which occurs in certain circumstances.

**Retroduction**
⇒ Intentional consideration, exploration and identification of the hidden elements of causation that lie beneath observable data and logic.

In preparation for the current Pathways Study described in this manuscript, a comprehensive realist synthesis of the literature was undertaken to provide an initial programme theory,[21] which was used to help guide the focus of scientific enquiry in the Pathways study, and inform the current protocol and study tools, through generation of an initial theory to be tested and further developed. The preparatory realist synthesis hence sought to answer: how, why, for whom, to what extent and in what contexts, do the social ties of hospital staff influence quality of care. Details of how the initial programme theory was developed can be found in the published realist synthesis—in brief, comprehensive literature searching, reviewing, data extraction, analysis and interpretation, was informed by engagement with stakeholders with relevant expertise and experience, from Sierra Leone and Kenya. The resultant theory comprised 35 context-mechanism-outcome configurations (CMOCs), organised under four emergent thematic domains: Social group, Hierarchy, Bridging distance and Discourse (see box 2). The Pathways Study will further develop and refine this initial programme theory for the specific setting of neonatal units in Kenya, using an open and explorative approach, cognisant that the majority of data used to produce the initial programme theory were from high-income settings.[21]

**Box 2    Summary of initial programme theory—taken from Blacklock *et al*[21]**

**Initial programme theory, to be tested (confirmed, refuted, refined) in the Pathways Study.**

Hospital staff prefer to communicate with colleagues who are similar to themselves, and with whom they share trust. However, this can create boundaries, silos and redundant information within pockets of the workforce, and different behavioural norms adopted by members of different groups, dominated by influential individuals. Tacit hierarchical rules also determine communication with others in the workplace based on status, forming a landscape of differential access to information and others. Fragmentation between different status groups can occur and space for status-enhancing behaviours is created. Workplace silos and status boundaries can be bridged in the presence of trust, confidence and mutual understanding, or where need is urgent and immediate. The organisation can actively comply with or challenge social boundaries within the workforce by the formal processes endorsed, which allow day-to-day performances of relative power, based on identity and hierarchical status. Attempts to manage the discourse, and the effectiveness of formal workplace communication processes, act to construct or disassemble social boundaries in hospitals. Connections between staff influence access to information and other forms of capital, behavioural norms and perceived agency. Ties are continuously shaped and as such are amenable to intervention. The capacity of these connections and the overall structure of the workplace social network are vital to delivering quality patient care.

The findings of the Pathways Study will make an important contribution to the ongoing work of KEMRI-Wellcome Trust Research Programme researchers in Kenya, particularly work focused on improvement, adoption of technologies and staff communication within the neonatal unit, and by exploring neonatal hospital care beyond the neonatal unit. The new understanding developed during the Pathways Study will be used for better designing and targeting interventions for quality improvement. Findings will also contribute to existing international literature on social networks of the healthcare workforce, the vast majority of which is derived from high-income settings.

### Aims and objectives

The aim of the Pathways Study is to explore how, why, for whom and in what circumstances, features of health systems 'software' (eg, values, norms, relationships) between health workers of all cadres caring for neonates in Kenyan hospitals, influence quality of care being targeted by improvement efforts. The specific objectives are:

I.    To describe how health workers of all cadres work together to deliver care to newborn babies in Kenyan hospitals, and the kinds of networks, relationships, social ties and personal/team-related sociocognitive skills that exist within and between the different groups of health workers caring for newborn babies.

II.    To examine how these relationships, networks, social ties and related sociocognitive skills influence nurses' clinical competencies in newborn care and

how these sociocognitive skills might be used to design recommendations to improve neonatal care in Kenyan hospitals.

## METHODS AND ANALYSIS

The Pathways Study is a realist evaluation,[20 22] which will employ a mixed-methods approach to collect relevant data, drawing on diverse methods.[23] Emergent semiregular patterns (ie, demi-regularities, box 1) and causative mechanisms will ultimately be identified from these data using a process of retroduction (box 1), to develop iterative explanatory theory for practical use. An initial programme theory derived from a realist synthesis of the literature[21] has been used to guide design of data collection tools for the Pathways Study and will also form a starting point for analysis and data-informed theory refinement. The first phase of data collection will be from two hospital case study sites, and from a 'nursing group' (phase 1, figure 1). A stakeholder codesign workshop will then be convened following the initial analysis, to further refine middle-range programme theory, and to develop recommendations for practice (phase 2, figure 1).

### Hospital case study sites

Data will be collected from two large urban public hospitals in Kenya. The study sites have been chosen in part due to ease of access, owing to the existing COVID-19 restrictions to movements. They are also facilities of contrasts with regard to setting (Hospital 1 and Hospital 2). The Pathways Study will collect data from eligible healthcare workers involved in the provision of care to neonates in the two hospital sites.

### Hospital 1

Hospital 1 is a large teaching and referral hospital, with a large neonatal unit with a newborn intensive care unit, and is an implementing site for ongoing initiatives to improve care of neonatal service through the 'NEST360,[5] Delivering and Sustaining Newborn Technologies' that seeks to integrate technological solutions within clinical care. The hospital is a training centre for basic diploma nursing and clinical medicine students, bachelor of science in nursing and bachelor of medicine and surgery undergraduate students, postgraduate diploma, higher diploma and master's level training for paediatric nursing, neonatal nursing, paediatrics and child health and a fellowship in neonatology.

### Hospital 2

Hospital 2 is a large county hospital, with a smaller neonatal unit. Unlike hospital 1, hospital 2 is a member of the 'Clinical Information Network' (CIN),[24] which seeks to improve clinical care of children by targeting better use of routine clinical data to inform policy and practice through providing audit and feedback mechanisms and supporting the work of clinical champions in participating network hospitals. Unlike hospital 1, hospital 2 is not an implementing site for NEST360.[5]

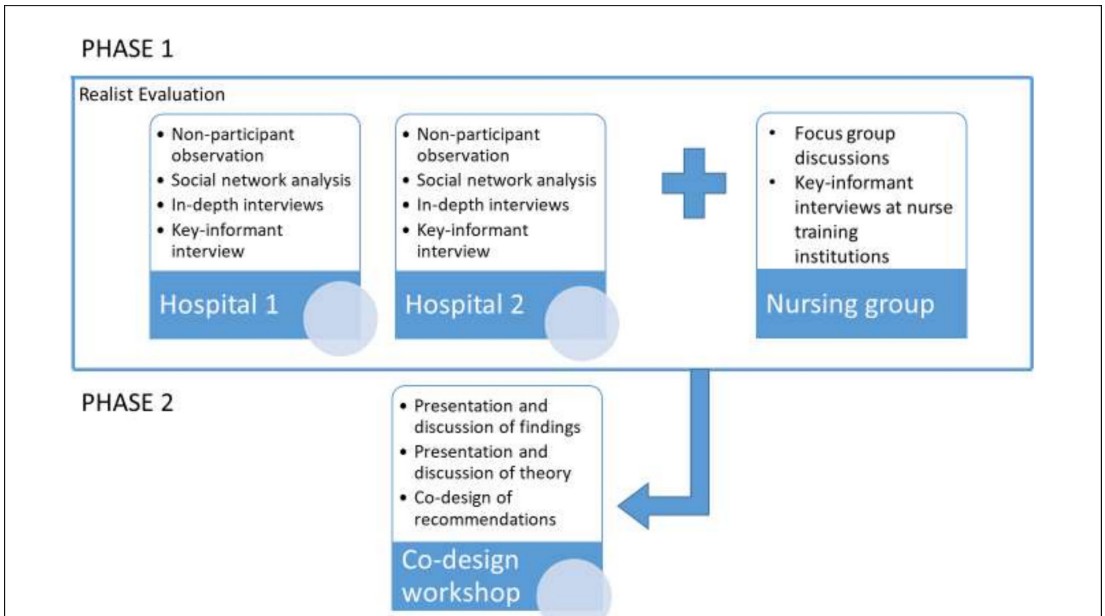

**Figure 1** A summary of mixed methods used in the Pathways Study. The figure describes the methodological approach for data collection in phase 1 (in the first three horizontal boxes), which involve data collection from the two case study hospitals and additional data from a nursing interest group, which will comprise nurses providing neonatal nursing care in the two case study hospitals and possibly other hospitals within Kenya. During this phase 1 work, we will also form a technical group for defining a realist evaluation programme theory. The last box, below the three boxes, describes a phase 2 data collection, where study findings from the phase 1 study participants as well as the initially developed programme theory, will be discussed over a 1-day workshop by a group of prior nominated stakeholders. These stakeholders will be drawn from memberships of neonatal nurses, neonatal and paediatric consultants, neonatal and paediatric professional associations, Kenya's Ministry of Health, division of newborn care, Kenya's medical and nursing practice regulatory agencies and senior health systems researchers from KEMRI-Wellcome Trust Research Programme. It is the recommendations from these stakeholder group that will inform the design of interventions targeting behaviour change among health workers providing care to neonates in Kenya.

### Nursing group

In addition to the hospital case studies, qualitative data relevant to exploring sociocognitive skills and the implementation of learnt knowledge and skills in the workplace, will be collected from a purposively sampled nursing graduate group and nurse educators from affiliated nurse training institutions.

### Stakeholder group

A group of 8–12 relevant stakeholders will be invited to participate in a co-production event, following initial data collection and analysis. Coproduced outputs will be refined theory and associated recommendations for practice. The invited stakeholders will be chosen from among the eligible groups shown in table 1, based on relevant knowledge and experience, and convenience.

Inclusion and exclusion criteria for participation are shown in table 1 below.

### Data collection procedures

We will conduct training for all researchers (including research assistants and study clinicians in the various study sites), including the requirements of their roles and how to collect the data using the study data collection tools (study tools attached as online supplemental appendix a-f). Development of the data collection tools (such as observation checklists, interview guides and social network questionnaire) was

informed by our previous realist synthesis of the literature.[21] These data collection tools will be piloted among the study researchers and colleagues from KEMRI-Wellcome whose work includes some engagements with CIN hospitals, and adapted prior to the start of data collection. To give feedback on the ongoing research process itself, including the piloting and adapting data collection tools, we will also convene a steering group of invited experts.

During data collection and analysis procedures, researchers will adopt a reflexive approach, that is, all members of the research team will be encouraged to keep aware of their own positionality and biases. Sampling will be purposive, seeking relevant data. The research team will intentionally seek, share and discuss potential biases throughout the course of the Pathways Study, and will mitigate as much as possible through openness and transparent and well documented processes. Training will help to improve individual self-awareness of researchers. The principal investigator and assistant research officer will meet regularly to compare and discuss emerging findings.

Data collection methods are listed below, with additional details provided in table 2.

### Non-participant observation

In the two hospital case study sites, we will observe health workers in the units/departments they are working.

**Table 1** Study inclusion and exclusion criteria

| Data collection | Inclusion criteria | Exclusion criteria |
|---|---|---|
| ▶ Non-participant observation<br>▶ Social network questionnaire<br>▶ In-depth interviews | ▶ Must work in the delivery room, postnatal ward, neonatal unit or paediatric ward, or support the function of these units as part of their work within the wider hospital.<br>▶ Relevant staff of all cadres and grades will be included: nurse, doctor, midwife, nutritionist, clinical officers, non-clinical and support staff, hospital administration, nursing, midwifery and final year medical and clinical officer students in the mentioned units<br>▶ Must be aged 18 years or over, be able to provide informed consent and available to participate in the study | ▶ Does not work in, or provide any support to neonatal care delivery in the hospital<br>▶ Aged under 18 years, unable to give informed consent or unable to participate in study, without participation adversely affecting care delivery |
| ▶ Key Informant Interviews | Senior nurse/manager in the hospital:<br>▶ Must be above 18 years and be able to provide informed consent<br>▶ Must be duly registered and licensed to practice at diploma in nursing and above<br>▶ Must be an employee of the hospital under study<br>▶ Must have formal leadership roles in the neonatal or paediatric ward<br>▶ Must be available to participate in the study, without their participation adversely affecting to the functioning of the neonatal unit | ▶ Junior nurses without leadership or management roles<br>▶ Nurses working on locum or other agencies<br>▶ Aged below 18 years, not willing or able to provide informed consent or unable to participate in the study due to work pressures. |
| | Nurse educator in the attached training facility:<br>▶ Must be above 18 years and be able to provide informed consent<br>▶ Must be duly registered and licensed to practice at diploma in nursing and above<br>▶ Must be an employee of the hospital under study<br>▶ Must have formal leadership roles in the neonatal or paediatric ward<br>▶ Must be available to participate in the study, without their participation adversely affecting to the functioning of the neonatal unit | ▶ Junior teaching faculty without leadership or management roles<br>▶ Educators working on part time or are employees of other agencies<br>▶ Aged below 18 years or not willing or able to give informed consent<br>▶ Unable to participate in the study due to work pressures. |
| ▶ Focus group discussions | Basic nursing graduates:<br>▶ Must have a diploma or degree in nursing and/or midwifery<br>▶ Must be registered and licensed by nursing council of Kenya to practice in Kenya<br>▶ Must be above 18 years and be able to provide informed consent<br>▶ Are working in hospitals in Kenya on full time, part time or locum basis<br>▶ Must be available to participate in the study | ▶ Nurses with certificate or postgraduate specialty nurse training<br>▶ Not willing or able to provide informed consent<br>▶ Unable to participate in the study due to work pressures. |
| | Nursing graduates: neonatal and paediatric nurses or maternal-neonatal nurses<br>▶ Must have specialty training in neonatal or paediatric or maternal-neonatal nursing and registered/licensed by Nursing Council of Kenya<br>▶ Must be above 18 years<br>▶ Must be able to provide informed consent<br>▶ Are working in hospitals in Kenya on full time, part time or locum basis<br>▶ Must be available to participate in the study | ▶ Nurses without specialty neonatal, paediatric or maternal-neonatal nursing<br>▶ Not willing or able to provide informed consent<br>▶ Unable to participate in the study due to work pressures. |
| | Nurse educators:<br>▶ Must have at a minimum, a postgraduate diploma in nursing/midwifery/medical education and registered and licensed by Nursing Council of Kenya<br>▶ Are working full time or part time at the schools of nursing for Kenya Medical Training College, Kenyatta National Hospital and University of Nairobi.<br>▶ Must be above 18 years, be able to provide informed consent and available to participate in the study | ▶ Nurses with certificate or diploma training<br>▶ Nurses working in other institutes other than study sites or without teaching roles<br>▶ Not willing or able to provide informed consent or to participate in the study due to work pressures. |
| ▶ Stakeholder workshop | ▶ Must have relevant experience in healthcare (paediatric/neonatal care) delivery, research or healthcare policy/education/decision making (health systems researcher, policy maker, health service leader, educator, professional representative or employee of national or county ministry of health or international health agency as assessed by KWTRP)<br>▶ Must be familiar with key aspects of intervention design or implementation (as assessed by colleagues in extended research team in Kenya) | ▶ Insufficient relevant experience or familiarity with key aspects of healthcare service delivery, research or policy/decision making (as assessed by KEMRI Wellcome Trust Research Programme, research colleagues) |

Sampling of shifts (six per hospital) will be discussed with the unit manager and shifts will be purposively chosen to select a range of different shifts and to capture variation. A sample of relevant hospital meetings will also be observed at each hospital site. Consent will be sought from staff at the start of the shift, and observations by the researcher will continue throughout the entire shift to try to mitigate any possible changes in staff behaviour due to being observed. Six shifts will be observed per hospital, to also try to mitigate any potential changes in staff behaviour due to being observed. Though not the focus of observations, caregivers will also have the presence of the researcher explained to them and be given

the option to opt out of them/their baby being present in observations of staff.

## Social network questionnaire

We will invite all health workers to participate (a complete network) who are involved in delivering neonatal care from relevant departments (ie, neonatal unit, paediatric and maternity wards, and the allied departments: laboratory, pharmacy, radiology, nutrition and patient support services). If the number of staff involved in delivering neonatal care exceeds 120, those staff who are most directly involved in the delivery of neonatal care by their place of work being on the neonatal unit, paediatric or

**Table 2** Data collection procedures

| Method | When | Average time taken | Justification |
|---|---|---|---|
| 1. Non-participant observation | Observations of 6 shifts per hospital site, as discussed with unit manager, to aim for maximum variation. A sample of hospital meetings will also be observed. | 6–8 hours per shift on average, plus 6–8 hours of meetings (total 42–56 hours of non-participant observation per site) | Observational data will be used to explore social ties and networks in the hospitals, and influence on provision of neonatal care<br>Objective 1 |
| 2. In-depth semistructured interviews | One interview per participant (up to total 35 in each of 2 hospital sites) | 45–60 min | In-depth interviews will explore the personal experience and reflections of staff as to how social ties are formed, with whom, and their role and influence in neonatal care provision<br>Objective 1 |
| 3. Social network questionnaire | One questionnaire per participant (total 40–120+ participants per hospital site, depending on size of networks) | 30–45 min | Social network data will help to explore the structure of staff communication networks and network characteristics<br>Objective 1 |
| 4. Key informant interview | One person per each hospital and one person per each attached training facility (total of 5–7 persons) | 30–45 min | Key informant interviews will provide additional contextual information<br>Objective 1 |
| 5. Focus group discussions | Three focus groups (each group of 5–9 members):<br>1. Basic nursing graduates (diploma and degree)<br>2. Specialist neonatal and paediatric nursing graduates<br>3. Nurse educators for both basic and specialised postbasic neonatal or paediatric nursing programmes. | Each focus group lasting 60–90 min | Focus group discussions will explore sociocognitive aspects of uing knowledge and skills in the workplace<br>Objective 1 |
| 6. Stakeholder Workshop | One session following the data analysis from methods 1–3.<br>Total of 8–12 participants | One day (either in-person or online) lasting 5–7 hours | The stakeholder workshop will provide data and use elements of codesign, to develop recommendations for how understanding of social ties could enhance efforts to improve quality of care for newborn babies<br>Objective 2 |

maternity wards, will be invited to participate in the social network questionnaire in the first instance.

A complete staffing list will be compiled by speaking to the management of the hospital/units and relevant student representatives (list will include names of staff of all cadres, students and all shifts). The researcher will assign a random ID code to each member of staff on the list, using a random sequence generator. Participants will be asked to identify colleagues they would speak to from a staffing list, and the researcher will use a separate research list to locate a corresponding ID code for each member of staff/student named during the social network questionnaire. Demographic data will be collected to identify and better understand homophily (those with similar demographics forming preferential ties with one another) within the network and where it occurs.[9] This will help to understand for example, whether cadre/gender/academic training is associated with connections in the workplace. If conducted online, participants will be asked to name colleagues without using the roster prompt, to avoid electronic sharing of staff lists.

### In-depth interviews
We will conduct purposive sampling to obtain data from approximately 35 study participants from each of the participating hospitals (total of approximately 70). We will seek variation of interview participants to identify and explore diversity of views and experience, for example, variation in cadres, units and demographics.

### Key informant interviews
One representative from management will be invited to participate in a key informant interview, from each of the two hospitals and from associated training institutions.

### Focus group discussions
Multistage sampling (convenience and snowballing then stratified sampling) will be used for focus group discussions of basic diploma and/or degree nursing graduates, specialist paediatric/neonatal nursing graduates and nursing educators. Participants from training institutions (nursing educators) will be recruited purposively. For nurse graduates, informal/formal professional nursing graduate social media group fora, such as WhatsApp and Telegram, will be used to invite individuals to participate in the study, followed by snowballing to reach those who are not on the social media platforms. This will be followed by stratification (1:1:1) to represent the various professional cadres, training programmes and facilities where they practice.

### Stakeholder workshop

We will conduct purposive sampling to obtain data from at least one representative among the various stakeholders, to achieve a sample size of 10–15.

### Analyses

The research objectives of the Pathways Study which aim to build theory, and the mixed-methods approach to data collection, make realist evaluation an appropriate scientific approach for analysis and interpretation.

A refined middle-range programme theory (box 1) and subsequent recommendations for practice will ultimately be codeveloped as the main outputs of the Pathways Study. The initial programme theory developed from literature (22, and box 2) will be refined and expanded through examination and analysis of data collected during the Pathways Study, by seeking out semi-regular patterns (demi-regularities, box 1) of outcomes that occur within the programme theory and then developing causal explanations for these outcomes in the form of CMOCs (box 1) using retroduction (box 1).[19 23] Hence, the initial programme theory derived from the literature will provide a series of causal explanations that can be 'tested' (confirmed, refuted or refined) against the primary data collected during the Pathways Study.[20] Where indicated by our interpretations of the primary data, new, expanded or revised CMOCs will be used to refine the initial programme theory for use in neonatal units in Kenya—thus further improving the level of understanding captured in the resultant middle-range programme theory (figure 2).

Prior to the application of the realist logic of analysis, we will do some additional data analysis, for example, thematically analysing the qualitative data and using social network approaches to analyse the quantitative data:

▶ Qualitative data from the study collected during non-participant observations, in-depth interviews and focus groups, and the stakeholder codesign event will initially be analysed thematically using NVivo software. This will lead to inductively developing a theoretical framework. Qualitative data on sociocognitive aspects of the study (key informant interviews and in-depth interviews) will be iteratively analysed using a grounded theory approach following an Open-coding →Axial-coding →Selective-coding approach to develop an explanatory theoretical framework on the role of sociocognitive skills in the utilisation of clinical competencies by nurses.

▶ Quantitative data collected from a Social Network Questionnaire will be analysed with standard social network analysis (SNA) metrics (eg, density, centrality) using Gephi free software and/or R to develop sociograms and a set of network measures.

Following the initial phase of realist anlysis and theory building (phase 1, figure 1), a group of stakeholders will be invited to review and further refine the theory, based on their expert knowledge and experiences (phase 2, figure 1). Through engagement with these stakeholders, explanatory theory will be further refined, and used to develop a set of recommendations for practice.

### Patient and public involvement

The Pathways Study research team includes a representative from each of the two hospital study sites and training institutions as study collaborators. The role of study site collaborators is to provide useful linkages for facility engagements before, during and after the study as well as facilitate the smooth conduct of the research by contributing to data collection through arranging for meetings with study participants. During the study, an advisory steering group will be formed of relevant coinvestigators who are not directly involved in data collection or ongoing analysis as well as extended colleagues working on relevant projects, to review study findings and provide

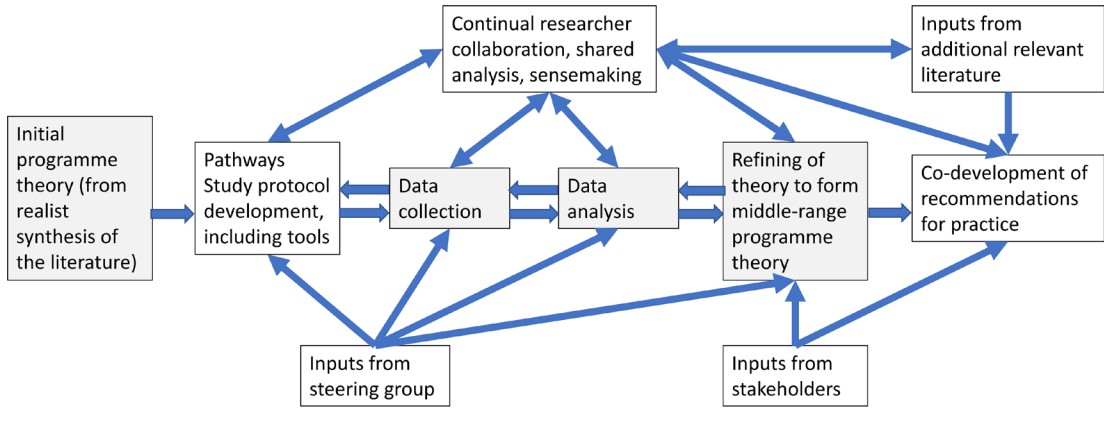

Note: Data collection and data analysis will be undertaken in an ongoing manner, so that findings can inform study tools to further confirm, refute or refine tentative emerging causal explanations.

**Figure 2** Programme Theory Development in the Pathways Study. The figure shows the dynamic and iterative process of theory development and refinement in the Pathways Study, starting from the initial programme theory, to the development of middle-range programme theory and recommendations for practice, using study data, steering group and stakeholder inputs, continual researcher collaboration and relevant literature.

useful comments and expert advice on study data analysis procedures and interpretation of the study findings.

## ETHICS AND DISSEMINATION

Ethical approval for the Pathways Study has been received from Kenya Medical Research Institute (KEMRI/SERU/CGMR-C/241/4374) and Oxford Tropical Research Ethics Committee (OxTREC 519-22).

Insights from the Pathways Study will add new understanding and a different theoretical lens to ongoing work by KEMRI-Wellcome Trust Research Programme in Kenya, particularly work focused on health workforce communication in neonatal care. Findings will also provide a useful addition to the existing body of literature on social networks of the hospital workforce, which is almost entirely derived from high income settings. The explanatory power of realist programme theory in specifically developing new understanding around the chains of causation associated with social ties of hospital staff, will be useful for designing and targeting interventions to enhance how such ties might positively contribute to change efforts in the neonatal unit. In addition, SNA is a relatively new methodology in healthcare research, and so the practical learning gained during the research process will be shared with KEMRI-Wellcome Trust Research Programme colleagues and others and will be useful for those intending to use SNA in future research.

Findings will further open-up the window of sociocognitive skills to policy makers and other stakeholders in Kenya, their role in service delivery quality and by way of interventions influence health workforce sociocognitive skills strengthening agendas. Findings relating to the role of non-technical sociocognitive skills in the use of learnt clinical competencies in neonatal care, will be used to develop recommendations for nurse educational interventions. Recommendations are likely to include how and when these skills can be incorporated into basic and specialised nursing programmes and in-service continuous professional development training activities. We anticipate that these findings might also be applicable to wider healthcare training programmes.

Findings will be shared with the two study hospitals, relevant educational institutions, and KEMRI-WELLCOME Trust Research Programme and the University of Oxford. Study findings will also be disseminated in seminars, local and international conferences, and as academic theses and research articles published in open-access scientific journals.

**Author affiliations**
[1]Health Systems and Research Ethics, KEMRI-Wellcome Trust Research Programme, Nairobi, Kenya
[2]Nuffield Department of Medicine, University of Oxford, Oxford, UK
[3]Nuffield Department of Medicine and Department of Paediatrics, Univerity of Oxford Nuffield Department of Medicine, Oxford, UK
[4]The Healthcare Improvement Studies Institute, University of Cambridge, Cambridge, UK
[5]Tropical Medicine, University of Oxford Nuffield Department of Medicine, Oxford, UK
[6]University of Oxford Nuffield Department of Clinical Medicine, Oxford, UK
[7]Centre for Geographic Medicine Research-Coast, KEMRI-Wellcome Trust Research Programme, Kilifi, Kenya
[8]Health Services Unit, KEMRI-Wellcome Trust Research Programme Nairobi, Nairobi, Kenya
[9]Nuffield Department of Primary Care Health Sciences, Oxford University, Oxford, UK

**Contributors** This protocol was codeveloped by CB and CW as joint authors/coprincipal investigators. ME, PMM, MB, GW and LH contributed to the development of the protocol, and JJ, SM and JJM contributed to the refining of the protocol and study tools for implementation.

**Funding** The Pathways Study is supported by the Wellcome Trust (#207522) through an award to ME as a Senior Fellowship. CB is studying for a DPhil—this work is supported by the Nuffield Department of Medicine, University of Oxford and the Medical Research Council (grant number MR/N013468/1).

**Disclaimer** The funders had no role in the preparation of this report or the decision to submit for publication.

**Competing interests** None declared.

**Patient and public involvement** Patients and/or the public were involved in the design, or conduct, or reporting, or dissemination plans of this research. Refer to the Methods section for further details.

**Patient consent for publication** Not applicable.

**Provenance and peer review** Not commissioned; externally peer reviewed.

**ORCID iDs**
Conrad Wanyama http://orcid.org/0000-0002-3553-6679
Mike English http://orcid.org/0000-0002-7427-0826
Lisa Hinton http://orcid.org/0000-0002-6082-3151
Peris Muoga Musitia http://orcid.org/0000-0003-4715-902X

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
