## [Reviewer comments · BMJ Open]

ARTICLE DETAILS

TITLE (PROVISIONAL)	Protocol for the Pathways Study: a realist evaluation of staff social ties and communication in the delivery of neonatal care in Kenya
AUTHORS	Wanyama, Conrad; Blacklock, Claire; Jepkosgei, Juliet; English, Mike; Hinton, Lisa; McKnight, Jacob; Molyneux, Sassy; Boga, Mwanamvua; Musitia, Peris; Wong, Geoff

VERSION 1 – REVIEW

REVIEWER	Tancred, Tara London School of Hygiene and Tropical Medicine, Disease Control
REVIEW RETURNED	23-Sep-2022

GENERAL COMMENTS	Many thanks to the authors for this very interesting protocol. I think it's written really clearly, but I do have a few concerns around clarity, especially in terms of theory development. I have more extensive comments for consideration below: - Rather than beginning the abstract with an indication of needing to meet SDGs, it might be more informative for readers to have a little bit more detail about why relational components among health workers are important.- Likewise, in the introduction, though a compelling case for a focus on relational aspects is made, it would be helpful perhaps to provide an example of what this means in more practical terms. It's not obvious to someone who's unfamiliar with the hardware/software framework and this conceptualisation of health systems functioning why relationship aspects might impact upon, for instance, clinical guidelines or decision making, so an example tying this together would be useful.- Prior to referring to mechanisms on line 23 (p.6), it might be useful to actually explain for the reader what these are and why they're linked to causality. Same thing for talking about "context-mechanism-outcome" configurations—I think this needs to be articulated more clearly for readers who are unfamiliar with realist evaluation (though who may be very interested in newborn care or relational aspects of care devliery).- Likewise, in line 32 (p.6), the "Pathways Study" is mentioned, but this is not something that the reader has been given information on. Though obvious, just signposting as the current study under description would be helpful.- I really like that this isn't heavy on realist jargon, but I'd perhaps add in brackets what you mean (e.g. calling your emergent theory from the realist synthesis an "initial programme theory").- Perhaps semantics, but on p.7, objective two reads a bit oddly with "use of nurses' clinical competencies"—the preceding content seems to suggest that these relational aspects can influence competencies (not just their use, but their development). Perhaps
---

	just remove “use of”?  - As a reader, I’d like to see more of your initial programme theory to better appreciate how you’re actually interrogating your CMOs. The data collection methods are detailed, but read rather like an exploratory qualitative study without linking this back to the purpose of realist inquiry. - See my comment below as I may have misunderstood this, but who was part of the development of (what I think was?) the initial programme theory following the realist synthesis? Was this totally an academic exercise, or were (which would be ideal) stakeholders engaged? - Please explain what’s meant by “they are also facilities of two contrasts” in the section for hospital case study sites. - Who are the members of the stakeholder group and why were they chosen? Or rather, who will they be if they are not already active? The inclusion criteria point to a broad range of possible participants. - Likewise who will be members of the steering group and how will they be identified? “Experts” is too vague. - On p.14, CIN hasn’t been actually used and defined (assuming it’s the Central Information Network, the acronym needs to appear on page 9). - Likewise, in reflecting on analysis, I think actually using terminology and description of retroductive analysis is really important. I think “realist logic” and how you plan to apply it needs to be more clear for the reader. - Under analysis, it’s a bit confusing to start by saying that a realist programme theory will be developed. It says this will be based on the realist review, but on p.6 it’s clearly indicated that a programme theory was generated (“the resultant programme theory”) from the realist review. It’s not then clear how the “realist programme theory” referred to on p.19 will be developed. By whom? When? Using what additional data sources? Are you suggesting that all the data collection will take place and this + realist synthesis findings will be used to develop an initial programme theory? Please clarify. - The stakeholder involvement mentioned on p.20—who are the stakeholders, and is this just to establish the “final” (mid-range) programme theory? It seems like they will input an initial programme theory, but this would not be where recommendations would be drawn from. Please clarify when you will engage them and to what end. As above, I think actually using some realist terminology (and introducing, perhaps in a box, what the terms mean) would be helpful. A figure to also walk the reader through the theory refinement process would be very useful. Figure 1 is helpful, but it’s a bit confusing, as it points to theory generation at only one point. I’d expect to see clearly where (and how) the initial programme theory will be developed, how it will be tested, how many iterations of the programme theory there will be, and how the “final” (mid-range) programme theory will come about. - To that end, how often will you be collecting/analysing data and testing your programme theory? Basically, how many iterations of theory development do you expect do you expect? If it’s basically just making an initial programme theory and then moving from that to the “final” programme theory, I would caution that this is not sufficiently robust.
--	--

REVIEWER	Degefie, Tedbabe Newborn and Child Health Consultant, UNICEF
REVIEW RETURNED	14-Dec-2022

GENERAL COMMENTS	It is well written protocol and addressing new emerging g area with
---

strong potential to inform human resource strategy.

VERSION 1 – AUTHOR RESPONSE

Reviewer: 1	
Many thanks to the authors for this very interesting protocol. I think it's written really clearly, but I do have a few concerns around clarity, especially in terms of theory development. I have more extensive comments for consideration below:	Thank you for your comments on this manuscript. We have tried to incorporate your helpful suggestions in the revised manuscript – please see below.
- Rather than beginning the abstract with an indication of needing to meet SDGs, it might be more informative for readers to have a little bit more detail about why relational components among health workers are important.	Thank you. We have revised the introduction of the abstract to now read: “The informal social ties that health workers form with their colleagues influence knowledge, skills and individual and group behaviours and norms in the workplace. However, improved understanding of these ‘software’ aspects of the workforce (e.g. relationships, norms, power) have been neglected in health systems research. In Kenya, neonatal mortality has lagged despite reductions in other age groups under 5yrs. A rich understanding of workforce social ties is likely to be valuable to inform behavioural change initiatives seeking to improve quality of care. This study aims to better understand the relational components among health workers in Kenyan neonatal care areas, and how such understanding might inform the design and implementation of quality improvement interventions targeting health workers’ behaviours.”
- Likewise, in the introduction, though a compelling case for a focus on relational aspects is made, it would be helpful perhaps to provide an example of what this means in more practical terms. It's not obvious to someone who's unfamiliar with the hardware/software framework and this conceptualisation of health systems functioning why relationship aspects might impact upon, for instance, clinical guidelines or decision making, so an example tying this together would be useful.	Thank you for this comment. In response we have tried to improve clarity by adding a definition of health systems software, as defined by Sheikh et al. “Software’ in health systems (in contrast to ‘hardware’, such as drugs, equipment, number of staff etc.) has been defined as “the ideas and interests, values and norms, and affinities and power that guide actions and underpin the relationships among system actors and elements”.(6)” and we have also expanded the example of Rogers' diffusion of innovation model. “The impact of relational ties on clinical decision making was demonstrated in the landmark study (15) showing profound peer influence on prescribing of a new drug by physicians in the United States of America. More recent studies have likewise/>> demonstrated peer group influence on both clinical practices and association with differences in patient outcome. (16-18)

- Prior to referring to mechanisms on line 23 (p.6), it might be useful to actually explain for the reader what these are and why they're linked to causality. Same thing for talking about "context-mechanism-outcome" configurations—I think this needs to be articulated more clearly for readers who are unfamiliar with realist evaluation (though who may be very interested in newborn care or relational aspects of care devliery).	Thank you. We have revised this section of the introduction to improve clarity, as well as other sections throughout the manuscript. "Furthermore, a detailed understanding of causation with a focus on explanatory 'mechanisms' (see supplementary Box 1: Definitions of Key Terms) will help to identify aspects of 'context' (Box 1) amenable to interventions that are likely to result in a desirable change in 'outcome' (Box 1), perhaps overlooked by more traditional research approaches. Through developing explanatory middle-range programme theory (supplementary Box 1), practical opportunities for improving software in neonatal care will be identified.(19,20)" As suggested we have also added Box 1: Definitions of Key Terms.   Context: A specific aspect of the setting/environment of study, which when present triggers the activation of a mechanism(s).   Mechanism: A latent (often invisible) property/entity, sensitive to variations in context, which when activated causes an outcome to occur.   Outcome: A desirable (or undesirable) event/occurrence, which is of interest by its presence or absence (or strength/degree of).   Context-mechanism-outcome configuration: A unit/statement/diagram which links and orders a context, mechanism and outcome, to provide a causal explanation. It is often abbreviated to CMO or CMOC.   Middle-range programme theory: The product of realist enquiry – an abstracted theory derived from relevant and robust data, which is both testable and transferable to other settings by application of abstracted learning.   Initial programme theory: A rough outline of a theory of causation pertaining to the question under study, to guide the process of investigation.   Demi-regularity: An observable pattern in the data, which occurs in certain circumstances.   Retroduction: Intentional consideration, exploration and identification of the hidden elements of causation that lie beneath observable data and logic.  	Context:	A specific aspect of the setting/environment of study, which when present triggers the activation of a mechanism(s).	Mechanism:	A latent (often invisible) property/entity, sensitive to variations in context, which when activated causes an outcome to occur.	Outcome:	A desirable (or undesirable) event/occurrence, which is of interest by its presence or absence (or strength/degree of).	Context-mechanism-outcome configuration:	A unit/statement/diagram which links and orders a context, mechanism and outcome, to provide a causal explanation. It is often abbreviated to CMO or CMOC.	Middle-range programme theory:	The product of realist enquiry – an abstracted theory derived from relevant and robust data, which is both testable and transferable to other settings by application of abstracted learning.	Initial programme theory:	A rough outline of a theory of causation pertaining to the question under study, to guide the process of investigation.	Demi-regularity:	An observable pattern in the data, which occurs in certain circumstances.	Retroduction:	Intentional consideration, exploration and identification of the hidden elements of causation that lie beneath observable data and logic.
Context:	A specific aspect of the setting/environment of study, which when present triggers the activation of a mechanism(s).																
Mechanism:	A latent (often invisible) property/entity, sensitive to variations in context, which when activated causes an outcome to occur.																
Outcome:	A desirable (or undesirable) event/occurrence, which is of interest by its presence or absence (or strength/degree of).																
Context-mechanism-outcome configuration:	A unit/statement/diagram which links and orders a context, mechanism and outcome, to provide a causal explanation. It is often abbreviated to CMO or CMOC.																
Middle-range programme theory:	The product of realist enquiry – an abstracted theory derived from relevant and robust data, which is both testable and transferable to other settings by application of abstracted learning.																
Initial programme theory:	A rough outline of a theory of causation pertaining to the question under study, to guide the process of investigation.																
Demi-regularity:	An observable pattern in the data, which occurs in certain circumstances.																
Retroduction:	Intentional consideration, exploration and identification of the hidden elements of causation that lie beneath observable data and logic.																
- Likewise, in line 32 (p.6), the "Pathways Study" is mentioned, but this is not something that the reader has been given information on. Though obvious, just signposting as the current study under description would be helpful.	Thank you for this helpful suggestion. We have amended the text to: "In preparation for the current Pathways Study described in this manuscript"																

- I really like that this isn't heavy on realist jargon, but I'd perhaps add in brackets what you mean (e.g. calling your emergent theory from the realist synthesis an "initial programme theory").	Thank you for this and similar comments regarding this aspect elsewhere in the manuscript. We have tried to improve the clarity around the initial programme theory and where this sits in the context of the Pathways Study. For example, we have changed the text here to: "In preparation for the current Pathways Study described in this manuscript, a comprehensive realist synthesis of the literature was undertaken to provide an initial programme theory (21) which was used to help guide the focus of scientific enquiry in the Pathways study, and inform the current protocol and study tools, through generation of an initial theory to be tested and further developed. The preparatory realist synthesis hence sought to answer: how, why, for whom, to what extent and in what contexts, do the social ties of hospital staff influence quality of care. Details of how the initial programme theory was developed can be found in the published realist synthesis – in brief, comprehensive literature searching, reviewing, data extraction, analysis and interpretation, was informed by engagement with stakeholders. The resultant theory comprised of 35 context-mechanism-outcome configurations, organised under four emergent thematic domains: Social group, Hierarchy, Bridging distance and Discourse (see Appendix for summary of initial programme theory). The Pathways Study will further develop and refine this initial programme theory for the specific setting of neonatal units in Kenya, using an open and explorative approach, cognisant that the majority of data used to produce the initial programme theory were from high-income settings. (21)" We have also added Figure 2 explaining theory development and have added a summary of the initial programme theory as a supplementary Box 2.
- Perhaps semantics, but on p.7, objective two reads a bit oddly with "use of nurses' clinical competencies"—the preceding content seems to suggest that these relational aspects can influence competencies (not just their use, but their development). Perhaps just remove "use of"?	Thank you. We have removed as suggested.
- As a reader, I'd like to see more of your initial programme theory to better appreciate how you're actually interrogating your CMOs. The data collection methods are detailed, but read rather like an exploratory qualitative study without linking this back to the purpose of realist inquiry.	Thank you again for your comments regarding the initial programme theory and its planned interrogation in the Pathways Study. In regards to the initial programme theory itself, we have provided more detail, with a summary provided as in supplementary Box 2 and cited the open access published review which will enable interested readers to access the full set of CMOs. In regards to improving the clarity of linking the methods of data collection to the intended realist inquiry, we have provided: Figure 2: Programme theory development in the Pathways Study and explanatory text, for example here:

	“A refined middle-range programme theory (Box 1) and subsequent recommendations for practice will ultimately be co-developed as the main outputs of the Pathways Study. The initial programme theory developed from literature (21, and supplementary Box 2) will be refined and expanded through examination and analysis of data collected during the Pathways Study, by seeking out semi-regular patterns (demi-regularities, Box 1) of outcomes that occur within the programme theory and then developing causal explanations for these outcomes in the form of context-mechanism-outcome configurations (CMOCs, Box 1) using retroduction (Box 1).(22) Hence, the initial programme theory derived from the literature will provide a series of causal explanations that can be ‘tested’ (confirmed, refuted or refined) against the primary data collected during the Pathways Study.(20) Where indicated by our interpretations of the primary data, new, expanded or revised CMOCs will be used to refine the initial programme theory for use in neonatal units in Kenya – thus further improving the level of understanding captured in the resultant middle-range programme theory. (Figure 2)”
- See my comment below as I may have misunderstood this, but who was part of the development of (what I think was?) the initial programme theory following the realist synthesis? Was this totally an academic exercise, or were (which would be ideal) stakeholders engaged?	Thank you for this comment. Stakeholders from Kenya and Sierra Leone were engaged during the realist synthesis process. This has now been specified in the text: “Details of how the initial programme theory was developed can be found in the published realist synthesis – in brief, comprehensive literature searching, reviewing, data extraction, analysis and interpretation, was informed by engagement with stakeholders with relevant expertise and experience, from Sierra Leone and Kenya
- Please explain what’s meant by “they are also facilities of two contrasts” in the section for hospital case study sites.	We have revised this sentence to read: “They are also facilities of contrasts with regards to setting (see below).”
- Who are the members of the stakeholder group and why were they chosen? Or rather, who will they be if they are not already active? The inclusion criteria point to a broad range of possible participants.	Thank you for this comment. We have revised the text to read: “A group of 8-12 relevant stakeholders will be invited to participate in a co-production event, following initial data collection and analysis. Co-produced outputs will be refined theory and associated recommendations for practice. The invited stakeholders will be chosen from amongst the eligible groups shown in Table 1, based on relevant knowledge and experience, and convenience.”
- Likewise who will be members of the steering group and how will they be identified? “Experts” is too vague.	Thank you. This has been amended to read: “During the study, an advisory steering group will be formed of relevant co-investigators who are not directly involved in data collection or ongoing analysis as well as extended colleagues working on relevant projects, to review study findings and provide useful comments and expert advice on study data analysis procedures and interpretation of the study findings.”
- On p.14, CIN hasn’t been actually used and defined (assuming it’s the Central	We have now clearly defined this acronym. Thank you.

Information Network, the acronym needs to appear on page 9).	
- Likewise, in reflecting on analysis, I think actually using terminology and description of retroductive analysis is really important. I think “realist logic” and how you plan to apply it needs to be more clear for the reader.	Thank you. We have now revised this text to improve clarity and have included retroduction, providing a definition of this in Box 1.
- Under analysis, it’s a bit confusing to start by saying that a realist programme theory will be developed. It says this will be based on the realist review, but on p.6 it’s clearly indicated that a programme theory was generated (“the resultant programme theory”) from the realist review. It’s not then clear how the “realist programme theory” referred to on p.19 will be developed. By whom? When? Using what additional data sources? Are you suggesting that all the data collection will take place and this + realist synthesis findings will be used to develop an initial programme theory? Please clarify.	Thank you. Please see responses above and associated revised text and additional Figure 2.
- The stakeholder involvement mentioned on p.20—who are the stakeholders, and is this just to establish the “final” (mid-range) programme theory? It seems like they will input an initial programme theory, but this would not be where recommendations would be drawn from. Please clarify when you will engage them and to what end. As above, I think actually using some realist terminology (and introducing, perhaps in a box, what the terms mean) would be helpful. A figure to also walk the reader through the theory refinement process would be very useful. Figure 1 is helpful, but it’s a bit confusing, as it points to theory generation at only one point. I’d expect to see clearly where (and how) the initial programme theory will be developed, how it will be tested, how many iterations of the programme theory there will be, and how the “final” (mid-range) programme theory will come about.	Thank you for this comment. We have tried to improve clarity by signposting to Figure 1 in this section of text. Please also see our responses above and the new Figure 2, to improve clarity around the process of theory development.
- To that end, how often will you be collecting/analysing data and testing your programme theory? Basically, how many iterations of theory development do you expect do you expect? If it’s basically just making an initial programme theory and then moving from that to the “final” programme theory, I would caution that this is not sufficiently robust.	Thank you for this comment. Please see the new Figure 2, which we have added to improve clarity around the process of theory development.
Reviewer: 2	
It is well written protocol and addressing new emerging g area with strong potential to inform human resource strategy.	Thank you.

VERSION 2 – REVIEW

REVIEWER	Degefie, Tedbabe Newborn and Child Health Consultant, UNICEF
REVIEW RETURNED	06-Feb-2023
GENERAL COMMENTS	Many thanks for this revised manuscript. I think it's reading much more clearly now (especially for novice realist evaluation-users!).